# The Impact of Bank Specific and Macro-Economic Factors on Non-Performing Loans in the Banking Sector: Evidence from an Emerging Economy

**Shakeel Ahmed** [1] , **M. Ejaz Majeed** [1], **Eleftherios Thalassinos** [2,*] and **Yannis Thalassinos** [3]

1 Department of Management Sciences, International Islamic University, Islamabad 44000, Pakistan; shakeelpccrwp@gmail.com (S.A.); mejazmajeed@yahoo.com (M.E.M.)
2 Faculty of Maritime and Industrial Studies, University of Piraeus, 185-33 Piraeus, Greece
3 Gulf University of Technology and Sciences of Kuwait, Mubarak Al-Abdullah Area/West Mishref 40006, Kuwait; thalassinos@hotmail.com
* Correspondence: thalassinos@ersj.eu

**Abstract:** The current study examines macro-economic and bank specific determinants of non-performing loans (NPLs) for commercial banks from 2008–2018. The Pakistani banking sector has observed a significant increase in NPLs. In addition, the current study is undertaken to fill this gap in the literature as most of the prior studies focus on the developed markets. In the current study, we prefer the system GMM estimator. Its reliability depends on the validity of the instruments. To testing the second-order serial correlation, we apply the J test for testing the validity of the instruments and the Arellano–Bond AR (2) test. Using dynamic-GMM estimations, we find that credit growth, net interest margin, loan loss provision, and bank diversification significantly increase NPLs, while operating efficiency, bank size, and ROA lower NPLs. In addition, higher interest rates, exchange rates, and political risk significantly increase NPLs, while GDP growth decreases NPLs. This paper provides a timely insight to management and policy makers about the determinants of NPLs. The findings help management to take corrective actions and policy makers may take into consideration the significance of macro-economic conditions while formulating policy regarding NPLs. Likewise, the study provides insight to potential investors to consider the findings while selecting better investment opportunity. The current study is the first of its kind focusing on the link among bank specific, macroeconomic variables, and non-performing loans within the specific context of an emerging economy, Pakistan.

**Keywords:** bank specific and macroeconomic variables; non-performing loans; GMM model

## 1. Introduction

Before financial crises hit the global economy in 2007–2008, we observed a relatively stable credit quality of loan portfolios across the world. Since then, there has been a sharp decline in bank asset quality because of the global economic recession. In this vein, the researchers used different alternative indicators as proxy for credit quality: non-performing loans (NPLs) (Fainstein and Novikov 2011; Pestova and Mamonov 2012) and loan loss reserves (Louzis et al. 2012). The application of these diverse approaches for exploring the credit quality (same problem) offers motivating likelihoods for new researchers in this field (Amin et al. 2019). In this study, we use NPLs as a proxy of credit quality of the loan portfolio. Over the last two decades, we found a consistency in defining NPLs. Finally, NPLs as a proxy of bad loans is based on the international accounting and financial reporting standards.

Literature shows that a rapid build-up of NPLs is one of the basic causes of banking crises (Amin et al. 2019; Ashraf and Butt 2019; Khan et al. 2020; Thalassinos and Stamatopoulos 2015; Thalassinos et al. 2015). In line with the argument, the world has

witnessed an onset of the global financial crisis in 2007–2008 since the levels of NPLs considerably augmented across countries. Though after global financial crises, the economies all over the world faced a rapid decline in credit quality, yet the impacts were significantly different for different groups of economies. Once the bank's asset quality is deteriorated, it may have contagion impacts on overall economy. As a result, the researchers in this field referred to NPLs as "financial pollution" since they are adversely associated with economic outcomes (Amin et al. 2019). Therefore, it is quite imperative to minimize NPLs because their minimal ratio ensures the restoration of a sounder banking system and fosters overall financial stability (Dimitrios et al. 2016; Ghosh 2015). It is necessary for banking regulatory authorities to introduce new regulation that may prevent any alarming increase in NPLs, and the State Bank of Pakistan (SBP) should act against those banks having high ratio of NPLs because all commercial banks are working under the supervision of SBP. However, any policy response by concerned authority in this context first require an in-depth understanding about the factors that cause NPLs in a specific economy. To address the concern, we examine both bank-level and macro-level factors that may have contributed (negative or positive) to NPLs in Pakistan. One of the basic reasons for conducting the study is a substantial increase in NPLs in Pakistan; NPLs rose to historic level of Rs 783 billion at the end of June 2019, mainly due to lower recoveries on the back of higher interest rate (State Bank of Pakistan 2019). Secondly, Pakistan is a bank-oriented economy and any negligence from state bank of Pakistan and other concerned authorities could be disastrous.

Based on the aforementioned facts, the current study provides a timely solution to the problem by exploring the determinants of NPLs in Pakistan. We include both bank-specific and macro-economic level factors that may have impacts on NPLs based on prior literature. The findings are not only useful for concerned authorities but also for stakeholders. The stakeholders may consider our findings while making their portfolio investments. On the other side, the timely study may help the concerned authorities to take corrective actions. This will help the policy makers to take essential measures to avoid any prospect crises that may otherwise be proved worsen. Furthermore, this study also adds to the existing literature in the context of emerging economy and findings may be useful for economies which are exposed to identical or similar problems.

The structure of the paper is as follows: Section 2 provides a brief literature review on both the macroeconomic and bank-level determinants of NPLs, and on empirical evidence related to the feedback effects of NPLs on the real economy. Sources of the data employed as well as the methodology are presented in Section 3. Section 4 shows the empirical results of determinants of NPLs. The next section concludes the paper and gives policy recommendations.

## 2. Literature Review

The literature on NPLs can be classified into different parts (Ghosh 2015). There are firm specific factors and macroeconomic factors that may have influence on NPLs (Ashraf and Butt 2019; Beck et al. 2015). The bank specific factors represent the factors that can be controlled by the bank itself or a central bank can introduce mechanisms to keep check on these factors (Beck et al. 2015; Louzis et al. 2012; Makri et al. 2014). On the other hand, macro-economic factors represent economic conditions that are often not controllable, and policy makers need to consider them while formulating their policies (Messai and Jouini 2013). Since we are interested only in highlighting the determinants of NPLs rather to make a review of the empirical literature, a short summary of related literature is if underlines the determinant of NPLs. At the first stage, the study highlights the bank specific determinants of NPLs.

### 2.1. Bank Specific Factors

#### 2.1.1. Credit Growth

Credit growth varies across banking sectors and its impacts are also significantly different. It has both effects on banks because of its features. It is considered as a good

indicator of the banking sector's stability (Jakubik and Moinescu 2015). Similarly, literature indicates that faster credit growth leads to the higher loan losses in USA (Makri et al. 2014). Loan loses increases with rise in supply of loans from banks by reducing interest rate and makes easer procedures of getting loans (Louzis et al. 2012). NPLs rise with the supply of loans in both developed and developing countries, resultantly performance of the banking sector declines. The banks with higher credit growth are more likely to expose themselves to NPLs specifically in developing economies where the returns are quite uncertain. Based on prior studies, following relationship is expected.

**Hypothesis 1 (H1a).** *Credit growth has a significant positive impact on NPLs in Pakistan.*

### 2.1.2. Loan Loss Provisions

Banks use loan loss provisions to cover different kinds of loan losses like NPLs, customer bankruptcy, etc. However, its minimum portion is consumed in NPLs as the banks have significant NPLs each year (Makri et al. 2014). Higher loan loss provision is an indication of management inefficiency and it often is positively associated with actual losses. Banks with poor credit quality are facing more risk in their loan portfolio that results higher NPLs (Beck et al. 2015). Based on literature, the following relationship is expected.

**Hypothesis 1 (H1b).** *Loan losses have a significant positive impact on NPLs in Pakistan.*

### 2.1.3. Bank Diversification

Banks have two major sources of incomes: interest and non-interest. Interest can be earned by different types of loans and investment securities, while non-interest earnings from asset management, fee and commission paying services, trading, and derivatives. In the modern banking system, non-interest income represents a vital source of diversification (Chen 2006). It also reduces the volatility of overall bank's income (Chiorazzo et al. 2008). Additionally, it enables banks to increase stockholders' worth by shifting emphasis from traditional income sources to non-interest income sources (Beck et al. 2015). At the same time, banks need to balance the mishmash of non-interest and interest income to reduce risk. Though non-interest income is an important source of diversification, yet contrary to conventional view; De Young and Roland (2001) pointed out that non-interest income increases the volatility of banks earnings. Based on these viewpoints, we are expecting the following relationship.

**Hypothesis 1 (H1c).** *Bank diversification has a significant negative impact on NPLs in Pakistan.*

### 2.1.4. Operating Efficiency

The effect of cost efficiency on NPLs is vague. On the one hand, banks spending less to monitor lending risks is considered more cost-efficient, but the possibility of increasing NPLs in future (Ozili 2019). It shows the negative effect of efficiency on NPLs. On the other hand, if banks devote higher costs to minimize lending risk, but NPLs cannot be control due to poor managerial skills, is described as bad management (Podpiera and Weill 2008). Similarly, Benthem (2017) research indicates an inverse relation between operating efficiency and NPLs. Based on prior studies, we are expecting the following relationship.

**Hypothesis 1 (H1d).** *Operating efficiency has a significant negative impact on NPLs in Pakistan.*

### 2.1.5. Bank Size

Bank size is considered as an important determinant of non-performing loans (Lu et al. 2005). There is mixed evidenced from literature about direction of relation between bank size and NPLs. Stern and Feldman (2004) identify that larger banks follow liberal credit policy, the chances of NPLs are higher as compare to smaller banks. On the other hand, some of the studies (Ozili 2019) find that larger banks have better management skills

to recover loans from borrowers, hence the negative impact of bank size on NPLs. Based on prior studies, we are expecting the following relationship.

**Hypothesis 1 (H1f).** *Bank size has a significant negative impact on NPLs in Pakistan.*

2.1.6. Bank Profitability

Maximization of profit is the core objective of commercial banks and it is observed that highly profitable banks have fewer chances to engage in high-risk activities; resultantly profitability has the inverse impact on NPLs (Gurbuz et al. 2013). Contrary to this, Bonin and Huang (2002) describe that credit policy is not only determined by earnings of banks, but it also effected by reputation of management and force to follow liberal credit policy, which shows a positive relation between profitability with NPLs. Similarly, Messai and Jouini (2013) shows negative relation between banks' profitability and NPLs. Based on prior studies, we are expecting the following relationship.

**Hypothesis 1 (H1g).** *Bank profitability has significant negative impact on NPLs in Pakistan.*

2.1.7. Net Interest Margin

Net interest margin is also an essential bank-specific determinant of NPLs. The latest research by (Bonin et al. 2005) asserts that net interest margin has a strong positive association with NPLs. Similarly, some other studies (Ozili 2019; Adusei 2018) argue that net interest margin has a positive association with the NPLs, because higher net interest margin increases the interest burden. Banks increase their interest margin to minimize their default risk, in this way a direct relationship is expected between interest margin and NPLs (Cavallo and Majnoni 2002). Based on the literature, the following relationship is expected.

**Hypothesis 1 (H1h).** *Net interest margin has a significant positive impact on NPLs in Pakistan.*

*2.2. Ownership Structure Factors*

2.2.1. Government Ownership

The financial institutions controlled by the government authorities are known as public banks. Performances of these banks are discussed with the help of three different views: social, political, and agency views. Public banks work for the economic and social welfare of society (Cotugno et al. 2013). They want to maximize the welfare of whole society. These banks also try to achieve their political goals by providing jobs and advancing loans to enterprises (Shleifer and Vishny 1994; Ahmed et al. 2020a). The agency views attached with social views and government owned banks are also helpful to maximize social welfare along economic development but may be involve in misallocation of resources (Beck et al. 2015). Hu et al. (2006) found that public sector banks have higher NPLs as compare to private sector banks in Taiwan. Based on the prior literature, the following relationship is expected.

**Hypothesis 2 (H2a).** *Government ownership has a significant positive impact on NPLs in Pakistan.*

2.2.2. Family Ownership

Family ownership is characterized as that the firms and banks controlled and managed by family members (Ozili 2019). There are two schools of thought regarding role of family ownership. First school declares that family ownership as a mechanism to reduce ratio of NPLs. The second school states that in family-owned banks, due to nepotism, credit quality cannot be assured. Most of the studies (Bonin et al. 2005; Ahmed et al. 2020b) evidenced better monitoring and managerial skills by family-owned banks as compared to government owned banks. The following relationship is expected based on empirical support.

**Hypothesis 2 (H2b).** *Family ownership has a significant negative impact on NPLs in Pakistan.*

### 2.3. Macro-Economic Factors

### 2.3.1. Interest Rate

Interest rate is one of the crucial determinants of NPLs. Rise in interest rates shows an increase in the volume of borrowers' debt and makes debt servicing more expensive and difficult, as a result NPLs expended (Boss et al. 2009). Contrary to these findings, NPLs diminish by reducing interest rate, and resultantly, bank's earnings also decrease. Similarly, P.K. Ozili and Outa (2017) evidenced the negative association between interest rate and NPLs in Malaysia. Siddiqui et al. (2012) find that instability in interest rate is mainly influencing on the volume of non-performing loans in Pakistani banking sector. The following relationship is expected based on empirical support.

**Hypothesis 3 (H3a).** *Interest rate has a significant positive impact on NPLs in Pakistan.*

### 2.3.2. Exchange Rate

The latest research conducted by Umar and Sun (2018) identify various crucial factors attached with non-performing loans and found that exchange rate is one of the important and significant determinants of NPLs. Similarly, Zia and Huma (2015) documented that exchange rate has statistically significant effects on NPLs. On the same lines, Khemraj and Pasha (2009) proved a strong association between effective exchange rate and NPLs. There are various responsible factors of loan losses in banking sectors of the USA, and exchange rate is one of them (Gambera 2000). Based on the literature, the following relationship is expected.

**Hypothesis 3 (H3b).** *Exchange rate has a significant positive impact on NPLs in Pakistan.*

### 2.3.3. Political Risk

The political situation of a country is a critical factor affecting on NPLs and working of banking system. Many studies such as (Boudriga et al. 2009) proved that organizational atmosphere and political uncertainty have a strong impact on ratio of NPLs. Similarly, Wheelock and Wilson (2000) pointed out that top management of banking sector is influenced by different social and political pressure groups especially in developing economies. In this situation, loans are not given on a merit basis, they are given on political basis. Ultimately, these loans are not utilized productively and not refunded properly. Creane et al. (2006) examined different political factors causing non-performing loans in various countries. He pointed out that weak judicial empowerment, bureaucracy, and poor implementation of credit policy are considered to be the most important factors of increasing volume of NPLs, and it is the main cause of banking failure. Based on the literature, the following relationship is expected.

**Hypothesis 3 (H3c).** *Political risk has a significant positive impact on NPLs in Pakistan.*

### 2.3.4. GDP Growth Rate

GDP growth rate is also a critical macroeconomic determinant of NPLs (Us 2017). Umar and Sun (2018) proved inverse relation between GDP growth rate and NPLs; shows higher economic growth will decrease the ratio of NPLs in the country. A similar relation between GDP growth and NPLs is obtained by Buncic and Melecky (2013). NPLs decrease with a rise in GDP growth and increase with a fall in GDP growth due to recession in economy (Fernández de Lis and Saurina 2000). The following relationship is expected based on prior studies.

**Hypothesis 3 (H3d).** *GDP growth rate has a significant negative impact on NPLs in Pakistan.*

## 3. Data and Methodology

### 3.1. Data Collection

In the current study, we used a balanced panel with 20 banks, some banks are owned by the government while others are working under private sector in Pakistan. In this way, determinants of NPLs in both sectors can be estimated. The data frequency is annual basis for 2006–2018. We obtained data from two main sources. First, we collected data for bank specific variables form the websites of State Bank of Pakistan and bankers' association of Pakistan. At the same time, we also cross-checked the data by matching with financial statement of each banks and missing data is also collected from financial statements. Our bank specific variables include equity to total assets ratio, ROA, ROE, and growth of gross loans. The other information (financial) is extracted from balance sheets, income statements and notes from the annual reports. Similarly, we derive macro-economic data from IMF websites. Each variable is defined in Table 1 below.

**Table 1.** Variables and their definition.

| Variable | Definition |
|---|---|
| **Bank specific factors** | |
| Credit growth | Loans-to-assets ratio |
| Loan loss provision | Ratio of loan loss provision provided by bank in its financial statement |
| Bank diversification | Ratio of non-interest income to total income |
| Operating efficiency | Ratio of non-interest expenses to total assets |
| Return on assets | Ratio of net pre-tax income to total assets |
| Net interest margin | Difference between lending and borrowing rate |
| **Ownership structure** | |
| Government ownership | Ratio of government ownership |
| Family ownership | Ratio of family ownership |
| **Macro-economic factors** | |
| Interest rate | Country interest rate specify by the State Bank of Pakistan |
| Exchange rate | Real exchange rate provided obtained from IMF website |
| Political risk | Political risk index constructed by PCA methodology |
| GDP growth | GDP growth rate obtained from IMF website |

### 3.2. Dependent Variable

In this analysis, we use the ratio of impaired (NPL) to total (gross) loans as our dependent variable. It is very important to mention that bank scope reports the level of 'impaired loans'. The reported impaired loan could be dissimilar from the official classification of NPLs. In comparison, the 'impaired loans' is an accounting concept, and it reflects cases in which it is likely that the creditor may not collect the full amount that is detailed in the loan agreement, whereas 'NPL' is a narrow concept, which predominantly reveals loans that are past due for more than 90 days. Acknowledging these differences, we treated 'impaired loans' as NPLs in this analysis. Based on these viewpoints, we use bank specific and macro-economic factors as determinants of NPLs in Pakistan.

### 3.3. Model Specification

The current study uses the GMM estimator. It is developed by Arellano and Bond (1991), Arellano and Bover (1995), and Blundell and Bond (1998). It is a dynamic panel data estimator, which is used in most of the studies in the recent past. The GMM estimator is most appropriate if (1) the number of periods is small and there is a presence of large number of individuals, (2) past realization is the dependent factors of dependent variable, (3) there is a probability of correlation between independent variables and contemporaneous or the past realizations of the error term, and (4) heteroskedasticity and autocorrelation exists in the individuals.

The literature highlights two types of GMM estimator. These are the Arellano–Bond estimator (difference GMM) and the Blundell–Bond estimator (system GMM). In estimation, the Arellano–Bond estimator converts all the regressors, applying differencing, and at that time applies the generalized method of moments. The Blundell–Bond estimator accepts that first differences of the instrument variables are orthogonal to distinct fixed effects. Hence, including this supposition deeply supports the efficiency of the estimator, because of the introduction of extra instruments. There are two equations in the Blundell–Bond estimator (the original equation and the transformed one). That is why it is called the system GMM estimator. In current study we prefer the system GMM estimator. Its reliability analytically depends on the validity of the instruments. For this purpose, to test the second-order serial correlation, we apply the J test of Hansen (1982) for testing the validity of the instruments, and the Arellano–Bond AR (2) test because once a lagged dependent variable is considered, a fixed effect model may be erroneous as the error term is expected to be correlated with the lagged NPLs term therefore resulting to inconsistent estimates. Hence, GMM is the most appropriate methodology for our sample size and another unique feature of dynamic panel model is that lagged dependent variables can be used to construct orthogonal conditions. As a result, GMM can be used to estimate the unknown parameters. The following equation is used for estimation:

$$
\begin{aligned}
NPLs_{it} = \alpha + \beta1 \quad & \text{Credit growth} + \beta2 \text{ Loan Loss provisions} + \ \beta3 \text{ Bank Diversification} \\
& + \beta4 \text{ Operating Efficiency} + \beta5 \text{ Bank Size} + \ \beta6 \text{ Bank profitability} \\
& + \beta7 \text{ Net interest margin} + \ \beta8 \text{ Government ownership} \\
& + \beta9 \text{ Family ownership} + \beta10 \text{ Interest rate} + \beta11 \text{ Exchange rate} \\
& + \ \beta12 \text{ Political risk} + \beta13 \text{ GDP growth rate} + \varepsilon
\end{aligned}
\tag{1}
$$

The above model consists of dependent variable (NPLs), independent variables (bank specific and macro-economic) and error term.

## 4. Empirical Results

Various statistical tests are applied to estimate the impact of different bank specific and macro-economic factors on NPLs.

### 4.1. Descriptive Statistics and VIF

The description of these variables is provided in Table 2 below. In descriptive statistics table, we provide the mean, median, and skewness only.

**Table 2.** Descriptive statistics.

|  | **Mean** | **Median** | **Skewness** |
|---|---|---|---|
| Credit growth | 0.655 | 0.603 | 4.452 |
| Loan loss provision | 0.051 | 0.048 | 3.293 |
| Bank diversification | 0.318 | 0.299 | 2.102 |
| Operating efficiency | 0.052 | 0.0531 | 4.304 |
| Bank size | 6.714 | 6.361 | 3.203 |
| Return on assets | 8.194 | 8.091 | 4.304 |
| Net interest margin | 0.061 | 0.059 | 1.592 |
| Government ownership | 0.993 | 0.905 | 3.203 |
| Family ownership | 0.410 | 0.408 | 2.192 |
| Interest rate | 7.19 | 7.08 | 4.364 |
| Exchange rate | 88.403 | 85.21 | 3.029 |
| GDP growth | 3.083 | 3.101 | 4.304 |
| Political risk | 53.450 | 53.426 | 4.666 |

We also applied the variance inflation factor (VIF) which is more appropriate to assess the multicollinearity. The values of all variables are below the benchmark 5.00, hence, there

is no possibility of a serious issue of multicollinearity in our sample size. The results are presented in Table 3 below.

**Table 3.** Variance inflation factor.

|  | VIF | Median |
|---|---|---|
| Credit growth | 2.4727 | 0.4044 |
| Loan loss provision | 2.1195 | 0.4718 |
| Bank diversification | 2.1195 | 0.4718 |
| Operating efficiency | 2.0606 | 0.4853 |
| Bank size | 2.0017 | 0.4996 |
| Return on assets | 1.8251 | 0.5479 |
| Net interest margin | 1.7270 | 0.5790 |
| Government ownership | 2.0042 | 0.4990 |
| Family ownership | 2.4051 | 0.4158 |
| Interest rate | 2.3382 | 0.4277 |
| Exchange rate | 2.2715 | 0.4402 |
| GDP growth | 2.0710 | 0.4829 |
| Political risk | 1.9597 | 0.5103 |

*4.2. Systems-GMM Estimation Results*

The results of the GMM estimator are presented in Table 4 below. For clarity, we used different models to justify our findings. In model 4, we regress the bank specific variable only. The findings show that credit growth is a positive and significant determinant of NPLs in Pakistan ($\beta = 0.194$ and $p < 0.05$: refer to model 4 Table 3). This is in line with the view that higher credit growth of the bank leads to market risk and the probability of loan default is much higher for these banks (Louzis et al. 2012; Makri et al. 2014). This is in line with earlier findings that reported positive association between credit growth and NPLs. Importantly, the findings also persist in other models with a decrease in coefficient value. Hence our Hypothesis 1a is supported. We also find a positive and significant association between loan loss provision and NPLs ($\beta = 0.092$ and $p < 0.05$: refer to model 4 Table 4). The findings support our Hypothesis 1b by suggesting that a higher loan loss provision is positively associated with NPLs. These findings are in line with earlier findings (Makri et al. 2014; Messai and Jouini 2013; Škarica 2014). Similarly, we find that greater diversification by banks is not statistically insignificant; thus, rejecting our Hypothesis 1c. The insignificant may be due to overall economic turndown in Pakistan. Furthermore, we find negative and significant association between bank operating efficiency and NPLs in Pakistan ($\beta = -0.167$ and $p < 0.01$: refer to model 4 Table 4). This shows that banks with higher operating efficiency are less likely to have higher NPLs ratio in Pakistan. Hence, our Hypothesis 1d is significantly supported in line with earlier findings of (Beck et al. 2013; Espinoza and Prasad 2010; Messai and Jouini 2013). Furthermore, we find negative association between bank size and NPLs ($\beta = -0.111$ and $p < 0.05$: refer to Model 4 Table 4). This indicates that banks with higher size are more likely to maintain their loan efficient and have better controlling mechanism (Espinoza and Prasad 2010). At the same time, these banks control the market and are better in managing risk than their counterparts. Importantly, we also find that profitable banks are less likely to have higher NPLs ration in Pakistan as findings show negative association between ROA and NPLs in Pakistan ($\beta = -0.172$ and $p < 0.05$: refer to Model 4 Table 4). This shows that profitable banks are efficient in cost and management and their loan exposure is comparatively secure which resultantly decreases the likelihood to have significantly higher NPLs. Lastly, we find positive and significant impacts of net interest margin (NIM) and NPLs in Pakistan ($\beta = 0.131$ and $p < 0.01$: refer to Model 4 Table 4). The banks with higher NIM expose themselves to market lending risk that results higher NPLs. This may be attributed to greedy lending that can be disastrous in future.

**Table 4.** Results of GMM estimations.

| | Model 1 | | Model 2 | | Model 3 | | Model 4 | |
|---|---|---|---|---|---|---|---|---|
| | Coefficient | t-Statistics | Coefficient | t-Statistics | Coefficient | t-Statistics | Coefficient | t-Statistics |
| Constant | −0.94 *** | −5.019 | −1.04 *** | 5.337 | −1.01 *** | 5.262 | −1.19 *** | 5.095 |
| NPLs (t-1) | 0.921 *** | 6.201 | 0.885 *** | 6.088 | 0.791 *** | 6.651 | 0.788 *** | 6.452 |
| Credit growth | 0.266 ** | 3.395 | | | | | 0.194 ** | 3.293 |
| Loan loss provision | 0.892 ** | 2.980 | | | | | 0.194 ** | 3.293 |
| Bank diversification | 0.0231 | 1.226 | | | | | 0.092 ** | 2.891 |
| Operating efficiency: | −0.198 *** | −5.665 | | | | | 0.0051 | 1.189 |
| Bank size | −0.091 ** | 2.465 | | | | | −0.16 *** | −5.49 |
| Return on assets | −0.219 ** | 3.085 | | | | | −0.111 ** | 2.391 |
| Net interest margin | 0.2019 *** | 4.528 | | | | | −0.172 ** | 2.992 |
| Ownership structure | | | | | | | | |
| Government ownership | | | 0.342 ** | 3.651 | | | 0.115 ** | 2.102 |
| Family ownership | | | −0.314 *** | −3.819 | | | −0.17 *** | −3.293 |
| Macro-economic factors | | | | | | | | |
| Interest rate | | | | | 0.287 ** | 3.299 | 0.216 ** | 3.20 |
| Exchange rate | | | | | 0.0128 ** | 2.052 | 0.037 ** | 1.99 |
| Political risk | | | | | 0.514 *** | 6.588 | 0.337 *** | 6.39 |
| GDP growth | | | | | −0.132 * | −1.815 | −0.012 * | −1.761 |
| Year effect | Yes | | Yes | | Yes | | Yes | |
| Chi sq | 255.001 | | 217.264 | | 381.316 | | 586.251 | |
| AR(1) (p-val) | −2.555 *** | | −3.562 *** | | −4.409 *** | | −4.167 *** | |
| AR(2) (p-val) | −0.271 | | −0.672 | | −0.458 | | −0.567 | |

\* Indicates significance at 10% level. \*\* Indicates significance at 5% level. \*\*\* Indicates significance at 1% level.

We include ownership structure variables in Model 4. It is important to mention that the level of significance for all bank specific variables remains same; however, we observe a minor increase/decrease in coefficient value which is quite negligible. However, the findings show that government ownership positively impacts NPLs in banking sector of Pakistan ($\beta$ = 0.131 and $p < 0.01$: refer to Model 4 Table 4). Higher government ownership leads to a higher NPL ratio. In Pakistan, historically government ownership is relatively involved in banking inefficiency in term of performance and loan management. Hypothesis 2a is supported. In contrast, the family ownership negatively and significantly impacts NPLs in Pakistan ($\beta$ = −0.17 and $p < 0.01$: refer to Model 4 Table 4). This implies that family-owned banks are better in risk management in line with earlier findings (Bonin et al. 2005; Ahmed et al. 2020b). Resultantly, we find significant support for our Hypothesis 2b.

In Model 3, we include macro-economic variables only. The findings show that higher interest rates lead to higher NPLs in Pakistan. This may be attributed to a significant increase in interest rates in the recent past. Hence, our Hypothesis 3a is supported ($\beta$ = 0.216 and $p < 0.05$: refer to Model 4 Table 4). Furthermore, we also find a positive and significant impact of exchange rate on NPLs. Pakistan is an import-oriented economy and any change in exchange rate significantly impacts on manufacturing sector resulting higher probability of default. Based on these findings, we find a significant support for our Hypothesis 3b ($\beta$ = 0.037 and $p < 0.05$: refer to Model 4 Table 4). In addition to this, our findings also depict positive and significant impacts of political risk on NPLs in Pakistan. Surprisingly, we find significantly higher coefficient value of political risk and level of significance in comparison to other macro-economic factors ($\beta$ = 0.337 and $p < 0.05$: refer to Model 4 Table 4), it shows that NPLs increases due to political influence on higher management of commercial banks, and it is more influential factor than other macro-economic factors. The coefficient value of political risk is higher than any of the bank specific factors. This implies that political risk is the most influential determinant of NPLs in Pakistan. However, this may be attributed to higher political risk in Pakistan because Pakistan is exposed to a higher political instability since its creation 1947. Hence, our Hypothesis 3c is strongly supported. Finally, we find a negative and significant association between GDP growth and NPLs in Pakistan. Higher GDP growth represents the health of the economy, GDP growth is more likely to decrease banks credit risk ($\beta$ = −0.012 and $p < 0.10$: refer to Model 4 in Table 4).

For clarity purpose, we regress each set of variables in different models (Models 1–3). It is worth mentioning that in Model 1, we find a significantly higher different coefficient

value bank specific determinant. These refer to endogeneity concerns because the coefficient value significantly decreases in Model 2 when we include ownership structure variables. Furthermore, in Model 4, it also drops due to inclusion of macro-economic factors in our main model. Similarly, the coefficient values and levels of ownership structure variable drops in Model 4 because of inclusion of macro-economic factors. The same scenario is observed for macro-economic factors.

We use the Arellano–Bond tests (AR (1) and AR (2)) to test the first and second order autocorrelation of the residuals. The results show that the AR (1) is significant at 1% level of confidence. It is important that AR (1) should reject the null hypothesis of no first order serial correlation, whereas AR (2) should not reject the null hypothesis of no second order serial correlation of the residuals. Hence, AR (1) and AR (2) are as per required estimation because AR (1) is significant at 1% while AR (2) is insignificant. This implies that our GMM estimations are consistent.

### 4.3. Comparative Analysis

For further analysis, we split the sample into small and large firms based on the average mean value of the sample size. In Table 5 below the findings are presented. The study compares small and large banks based on certain reasons. First, large banks offer a highly inclusive list of financial accounts and services even from deposit accounts to loans to investing to wealth. Bigger banks tend to put in the investment to adopt and pioneer new financial technologies that can improve the service provided to customers. In contrast, the smaller banks seem to be more efficient and cost effective. For this purpose, we regress two different models; Model 1 presents the findings of small banks sample while Model 2 depicts the results of large banks. In Model 3, we included the findings of our main model from Table 3. Model 3 is our baseline regression results. In comparison, we find a lower coefficient value and level of significance between credit growth and NPLs in the case of larger banks ($\beta = 0.085$ and $p < 0.10$: refer to model 1 in Table 5). In contrast, the coefficient value and level of significant association between credit growth and NPLs for smaller banks is much higher in comparison to Model 3 ($\beta = 0.085$ and $p < 0.10$: refer to Model 2 in Table 5). This implies that larger banks are comparatively better at managing their credit growth as there is a weaker relation between credit growth and NPLs for them ($p < 0.10$). Other variables of interest that are significantly different for larger banks are bank diversification, operating efficiency, and ROA. We find bank diversification has comparatively less coefficient value and level of significant with NPLs ($\beta = 0.008$ and $p < 0.10$: refer to model 1 in Table 5). Again, this implies that larger banks are comparatively better in asset diversification. However, we find no significant difference for small banks in terms of coefficient and level of significance. Furthermore, we find a negative and significant impact of operating efficient on NPLs in Model 1 ($\beta = -0.0854$ and $p < 0.05$: refer to Model 1 in Table 5). In Models 2 and 3, the relation is insignificant. Here again, we find that larger banks are better in operating efficiency in comparison to their counterpart.

Among the corporate governance factors, we find no difference in context of family and government ownership for small and larger banks. The results remain almost quite like Model 3 for both groups, and we also observe no significant change in coefficient value. In addition, the results for association between macro-economic factors and NPLs are significantly different for political risk and interest rate. We find a higher coefficient value and level of significance for smaller banks in Model 2 ($\beta = 0.301$ and $p < 0.01$: refer to Model 2 in Table 5). In contrast, the results show a significantly weaker association between interest rate and NPLs in model 1 ($\beta = 0.115$ and $p < 0.10$: refer to model 1 in Table 5). This implies that larger banks have more capacity to deal with market interest rate as these banks are efficient in management. Similarly, their control over markets enables them to change according to market interest rate and furthermore, their better management also empowers them to collect more loans from the market. The positive role of political risk is more pronounced in Model 1 (for higher banks) ($\beta = 0.417$ and $p < 0.01$: refer to Model

1 in Table 5). This shows that larger banks are more exposed to political risk due to their more investment in manufacturing sectors. Political risk significantly impacts the overall performance of economy. Higher market exposure makes larger banks investments more volatile in comparison to their smaller counterpart. In smaller banks, the role of political risk is significantly lower implying that these banks are less exposed to political risk. Lastly, we do not find any significant change in impacts of exchange rate and GDP growth in Models 1 and 2.

**Table 5.** Comparative analysis.

| | Model 1 | Model 2 | Model 3 |
|---|---|---|---|
| | Coefficient | Coefficient | Coefficient |
| Constant | −1.56 *** | −1.086 *** | −1.19 *** |
| NPLs (t-1) | 0.298 *** | 0.541 *** | 0.788 *** |
| Bank specific factors | | | |
| Credit growth | 0.085 * | 0.214 ** | 0.194 ** |
| Loan loss provision | 0.168 ** | 0.192 ** | 0.194 ** |
| Bank diversification | 0.0080 | 0.102 ** | 0.092 ** |
| Operating efficiency | −0.0854 ** | 0.0011 | 0.0051 |
| Return on assets | −0.111 ** | −0.111 ** | −0.111 ** |
| Net interest margin | −0.172 ** | −0.172 ** | −0.172 ** |
| Ownership structure | | | |
| Government ownership | 0.100 ** | 0.122 ** | 0.115 ** |
| Family ownership | −0.21 *** | −0.15 *** | −0.17 *** |
| Macro-economic factors | | | |
| Interest rate | 0.115 * | 0.301 *** | 0.216 ** |
| Exchange rate | 0.033 ** | 0.020 ** | 0.037 ** |
| Political risk | 0.417 *** | 0.017 * | 0.337 *** |
| GDP growth | 0.014 * | 0.011 * | 0.012 * |
| Year effect | Yes | Yes | Yes |
| Chi sq | 661.178 | 476.109 | 586.251 |
| AR(1) (*p*-val) | −4.172 *** | −2.187 *** | −4.167 *** |
| AR(2) (*p*-val) | −0.648 | −0.546 | −0.567 |

\* Indicates significance at 10% level. \*\* Indicates significance at 5% level. \*\*\* Indicates significance at 1% level.

## 5. Conclusions, Limitations, and Future Recommendation

The study explores some of the key factors affecting non-performing loans in banking sector of Pakistan. The findings depict that credit growth; loan loss provision and bank diversification are positive and significant determinants NPLs in Pakistan. Based on these findings, we conclude that greater credit growth may be helpful to safeguard more profits, but it augments NPLs. Hence, there should have been an optimal extent of capital in banks' balance sheets along with maintaining high credit standards to lessen NPLs while nourishing ROA and better credit worthiness. Similarly, better quality management in terms of diversification is required to minimize the NPL ratio. The findings also show that efficient cost management reduces the probability of high NPLs ratio. Therefore, it is mandatory for banks to have effective cost management as it is a prerequisite to lessen NPLs and enhance the quality of balance sheet. The bank's profitability is also one of the important determinants of that reduces NPLs. More profitable banks have more prudent market lending, and they are financially stable at the same time. It is a time to reduce government ownership in the Pakistani banking sector as it proves to be costly in terms of NPLs and government owned banks are more influenced by political factors. Higher government ownership proves to be costly, and it may result in higher banking inefficiency. In contrast, family ownership leads to better credit management and improved bank's efficiency. Furthermore, GDP growth significantly affects NPLs, thus highlighting its countercyclical nature. Any improvement in GDP is vital to reducing NPLs. In contrast, higher interest, and exchange rates augment NPLs. These variables counter the countercyclical nature of GDP. However, the political risk is the positive determinant

of NPLs in Pakistan. Banks need to consider political uncertainty while formulating their loan policies. In comparison, we find larger banks are efficient in management, better in addressing the economic concerns and have higher profitability. This significantly reduces their NPLs ratio in Pakistan.

Findings of the study are helpful for management of banking sector to revisit lending policy and highly observe that loans cannot be used for non-productive purpose. To maintain stability of banking sector, higher authorities must establish more effective monitoring process to reduce the ratio of non-performing loans in Pakistani banking industry. It provides guideline to higher authorities of government to control corruption and made accountability process more effective along with no political involvement in banking sector. The comprehensive and effective governance system is required to minimize the volume of NPLs in Pakistan; these findings are beneficial for mangers of banks for policy making.

Furthermore, the study has various dimensions of contributions but on the other hand, it has few limitations. The study examines the impact of bank specific and macroeconomic factors on NPLs by using secondary data rather than primary data. Another limitation is related to data availability by commercial banks. There were hardly 20 commercial banks operating in Pakistan as of the beginning of 2000. Some other bank specific factors like profitability, size of liquidity, and macroeconomic factors—such as unemployment and inflation—should be included in future research.

**Author Contributions:** The Conceptualization and the initial idea was presented by S.A. and M.E.M., however E.T. and Y.T. finalize it and made in practical. The methodology and data analysis are performed by S.A. and M.E.M., while proof reading and verification of results is accomplished by E.T. Y.T. is responding for the overview of the article, corrections, and result clarifications. All authors have read and agreed to the published version of the manuscript.

**Funding:** This research work was not funded by any organization.

**Institutional Review Board Statement:** Not applicable.

**Informed Consent Statement:** Not applicable.

**Data Availability Statement:** We used a balanced panel with 20 banks, some banks are owned by the government while others are working under the private sector in Pakistan. In this way, determinants of NPLs in both sectors can be estimated. The data frequency is in an annual basis for the period 2006–2018. We obtained data from two main sources. First, we collected data for bank specific variables form the websites of State Bank of Pakistan and bankers' association of Pakistan. At the same time, we also cross-checked the data by matching with financial statement of each bank while other missing data were collected from financial statements. Our bank specific variables include equity to total assets ratio, ROA, ROE, and growth of gross loans. The other information (financial) is extracted from balance sheets, income statements and notes from the annual reports. Similarly, we derive macro-economic data from IMF websites.

**Conflicts of Interest:** The authors declare no conflict of interest.

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
