# Peer review of "The Impact of Bank Specific and Macro-Economic Factors on Non-Performing Loans in the Banking Sector: Evidence from an Emerging Economy"

_jrfm, doi:10.3390/jrfm14050217_

Round 1

Reviewer 1 Report

The paper examines the determinants of non-performing loans in a sample of 20 banks in Pakistan over the period from 2008 to 2018. It considers bank-specific as well as macroeconomic factors. 

  1. The estimation model is introduced in the equation on p. 8. However, the symbols are never defined. Even without knowing the meaning of the symbols, it is evident that the formula is wrong. For example, the first sum runs from J=1 to J, which does not make sense. The second sum uses index J (J=1 to k), but this index is never used in the sum’s expression. Finding such errors in the only equation of the paper calls into question whether the estimation (which is quite demanding) has been correctly implemented. Therefore, this negligence would justify a “Reject” recommendation.
  2. When presenting the method used in this paper, Roodman (2009) demands: “Report all specification choices. Using these estimators involves many choices, and researchers should report the ones they make—difference or system GMM; first differences or orthogonal deviations; one- or two-step estimation; nonrobust, cluster–robust, or Windmeijer-corrected cluster–robust errors; and the choice of instrumenting variables and lags used.” This is a meaningful recommendation that is not met in this paper. The indications on the specifics of implementing the system-GMM method are not detailed enough.
    Reference: Roodman, D. (2009): How to do xtabond2: An introduction to difference and system GMM in Stata, in: The Stata Journal 9 (1), 86-136.
  3. The Section on “Model Specification” starts with: “The current study uses the GMM estimator.” However, the model (in this case, a dynamic panel model) has to be introduced before the estimator (based on this model) can be specified. Several times in the text, the GMM estimation is treated as if it were the model to be estimated.
  4. The use of Loan loss provision as a determinant of non-performing loans seems to be tautological because the impairment of non-performing loans is technically achieved by recognizing a loan loss provision.
  5. More information should be given on the importance of the sample banks within the banking system in Pakistan.

Author Response

Dear Colleague,

Thank you for your valuable comments. Please find attached the corrected version of our manuscript.

Regards,

E. Thalassinos

Reviewer 2 Report

Comments for authors

JRFM 1191864

The Impact of Bank Specific and Macro-Economic Factors on Non-Performing Loans in Banking Sector: An Evidence from Emerging Economy

General comments:

The paper investigates the features of nonperforming loans in the Pakistani economy. The authors use the GMM estimator by Arellano Bond (1991) and employ company-specific (bank-specific) features and macroeconomics indicators to assess the situation. For analysis, the paper uses data following the period of the financial crisis. The study wants to help the bank management to take corrective decisions concerning the management of banking institutions. The study is interesting and timely.

  1. Introduction

This paper provides timely insight for management and policymakers about the determinants of non-performing loans in Pakistan. The motivation is coming from the fact that in Pakistan there was: “a substantial increase in NPLs in Pakistan; NPLs rose to historic level of Rs 783 billion at the end of June 2019, mainly due to lower recoveries on the back of higher interest rate (state bank of Pakistan report 2019). Secondly, Pakistan is a bank-oriented economy and any negligence from concerned authorities could be disastrous.” Page 2.

The introduction is well written and describes the motivation for the paper well.

  1. Literature Review

The literature review is organized systematically, and it summarizes relevant studies concerning non-performing loans and their features. LR is logically structured into sections on bank-specific factors, ownership factors, and macroeconomic factors. I think it would help if the authors added more information on bank regulation and capital adequacy measures in the country. It would be beneficial if the authors could explain how the measures are practised in Pakistan. 

  1. Data and Methodology

Data and methodology are characterised in section 3. The model is well specified and the applied methodology is suitable.  Data is covering the appropriate period (2006-2018), enabling the analysis.

  1. Empirical Results

Part 4.2, shows the results of GMM Estimation.  

Models 1-3 demonstrate the following:

Firstly, the results show that non-performing loans are positively related to bank-specific factors: non-performing loans from the previous year, credit growth, loan loss provisions, net interest margin, and negatively related to operating efficiency, banks size, and ROA. 

Concerning the ownership structure: both ownership structure factors are significant - family ownership and state ownership. Family ownership is negatively related to NPLs. 

Concerning macroeconomic factors: all macroeconomic factors are significant, although GDP growth is negatively significant, whereas interest rate, exchange rate, and political risk are positively significant. In particular, political risk is the most significant (can the authors explain, how was the political risk measured and what are the particular features of this factor in Pakistan?)

Model 4:

In the fourth model, the authors show differences between smaller banks and bigger banks and provide useful specific findings as to the structure of the impact of factors on the bank sector. Sensitivity to political risk is more significant in bigger banks.

Comparative analysis between the bigger and smaller banks in Model 4 brings useful perspective and shows different sensitivity of particular features when separating Pakistani banks according to their size.

  1. Conclusion, limitations, and future recommendations

The paper is well written.

Please explain in mode details how does the bank supervision work in Pakistan and what are the measures to prevent the existence and spreading of non-performing loans in the system. Please return to political factors and provide more explanation of the probable causes of your findings.

All in all, the paper deals with an important and timely topic, it is well written and interesting. Please, explain the background of bank regulation in Pakistan in more detail and also provide reasons for the major significance of political risk coming from your analysis. Minor English editing is required.

Author Response

Dear Colleague,

We thank you for your valuable comments. We have adjusted our manuscript accordingly. 

Regards,

E. Thalassinos
